# Teaching Bilingually: Unlocking the Academic and Cognitive Potential—Teachers' Insights

Emma Creed [1], Roberto Filippi [2] and Andrew Holliman [2,*]

[1] Department of Psychology, Arden University, Coventry CV3 4FJ, UK; emmacreed@googlemail.com
[2] Department of Psychology and Human Development, Institute of Education, University College London, London WC1H 0AA, UK; r.filippi@ucl.ac.uk
* Correspondence: a.holliman@ucl.ac.uk

**Abstract:** Recent quantitative research on bilingual education suggests beneficial academic outcomes for pupils regardless of socio-economic status (SES). Bilingual education in England, a relatively novel phenomenon, may be better understood from teachers' perspectives; however, there is a paucity of qualitative research in this area. To fill the gap, the present study explores the unique perspective of a sample of bilingual schoolteachers regarding the children who attend bilingual schools and the effects of bilingual education on children's academic and non-academic development. Semi-structured interviews were conducted with six bilingual schoolteachers, and thematic analysis was adopted to interpret their experiences in greater depth. A thematic analysis revealed four superordinate themes: Academic and Socio-Cultural Effects, Privilege in Bilingual Education, Bilingual Education Takes Time, and Special Educational Needs. These results highlight academic, socio-cultural, and linguistic benefits with important implications for promoting equity in bilingual education. Further research should focus on schools with a higher proportion of students from lower socio-economic status backgrounds.

**Keywords:** bilingual; school; teacher; thematic analysis

## 1. Introduction

Modern foreign language (MFL) teaching in the UK at the end of the twentieth century has undergone vast changes. Peaking in 1997 with language GCSE (General Certificate of Secondary Education) applications at 73% for girls and 82% for boys [1]. A mere seven years later, following the declassification of languages as compulsory at GCSE level, numbers declined to just 47% of GCSE applicants that year. The current national curriculum includes compulsory language teaching for children in key stage two (KS2) [2]. However, even Ofsted's (Office for Standards in Education) own review in 2021 (Ofsted, 2021) [3] highlighted ineffective and inconsistent provision across settings. Finch et al. [4] argued that despite the consistent increase in linguistic diversity nationwide, delivery of MFL was erratic. For smaller countries within the British Isles, some schools have been working towards bilingual outcomes for some time and for various reasons. Bilingual schools are on the rise in Wales, Ireland, and even the Isle of Man as a means of preserving endangered languages [5–8]. In England, whilst a few schools have employed bilingual methodologies such as intensive immersion courses, Content and Language Integrated Learning (CLIL), and a range of immersion programmes with varying schedules, until 2012, these were purely independent schools. England's state school system is overwhelmingly monolingual despite approximately one-fifth of primary age children having English as an additional language (EAL) [9]. In 2010, the UK government initiated the "free" school scheme, granting newly established schools more independence in terms of curriculum and vision funded directly from central government removed from local authorities [10]. Consequently, small, committed groups began establishing bilingual free schools. Bilingual education as a

pedagogy is distinct from educating bilingual children or EAL children. Genesee (2004) [6] explained it as a method of academic instruction that advances linguistic competence. Teaching EAL children, on the other hand, involves initially instructing them in their native language and gradually transitioning to the majority language to aid integration and cultural cohesion. This also more typically happens with individuals or smaller groups of children to aid assimilation with their peers. Conversely, educating children bilingually involves pedagogies such as 50:50 immersion, using immersive teaching to contextualise their learning through the target language. Unlike EAL teaching, this will typically apply to larger groups, such as an entire class or school. Baker and Wright (2017, pp. 371–380) [5] explained three perspectives on understanding bilingualism: as a problem, a right, or a resource. Prejudice towards linguistic minorities and political considerations contribute to the perception of bilingualism as a problem. However, Lamb et al. (2020) [11] argued for a more outward-looking approach, challenging monolingual frameworks and working towards improving social justice, and Charoenphon (2023) [12] makes a strong case for greater cultural awareness, social integration, and self-esteem as improved secondary outcomes for minority students working towards creating understanding of bilingualism as a right. Furthermore, Skinner and Meltzoff (2018) [13] found that imagined contact with members of different groups and structured intergroup contact were two key factors in reducing childhood intergroup bias. This highlights how bilingualism can be a resource in reducing prejudice and working towards improving social cohesion.

Bilingual education is a complex phenomenon with a simple label [5] (Baker & Wright, 2017, p. 197) and it is important to distinguish for clarity. Different pedagogies are employed according to their target audience and desirable outcomes. Baker and Wright (2017) [5] address these forms, categorising them into monolingual, weak, and strong forms according to the typical child in that education, classroom language, societal linguistic, and literacy aims. Submersion education, for example, for a language minority child (such as EAL children), has the goal of assimilation and the outcome of monolingualism and monoliteracy. Children placed in such education often encounter a preference for the majority language and negative emotions surrounding their heritage language [14]. In contrast, a 50:50 immersion methodology is considered a strong form of bilingual education [5]. This model involves teaching 50% in language one (L1, usually the dominant or official language of the country) and 50% in a second language (L2). That division can be divided by time or by curriculum. Evidence suggests positive outcomes for children in 50:50 immersion programmes [15–18]. Watzinger-Tharp et al. (2015) [18] expanded on earlier research, finding significant results in academic outcomes with students enrolled in dual language immersion programmes making greater mathematical achievements than their monolingual peers. Serafini (2019) [17] revealed superior academic benefits across all measures for fifth-grade children, although it was highlighted that those bilingual models that supported heritage language and culture resulted in greater growth. Bialystok (2018) [15] paid attention to SES and concluded net benefits across many domains, including bilingualism and biliteracy. Furthermore, Chamorro and Janke (2020) [16] measured significantly higher outcomes in cognitive tests measuring selective attention and response inhibition, and socially, in communication and cooperation skills for previously monolingual children who had spent only one year in bilingual education when compared to those in monolingual education.

Meanwhile, other studies have considered factors such as socio-economic status (SES), such as Lorenzo et al. (2020) [19], whose research on equity in bilingual education revealed encouraging results for students regardless of their SES; however, the majority of these studies have utilised quantitative research methodologies. The dearth of qualitative research conducted in this domain has resulted in the absence of adequately represented viewpoints of educators pertaining to student outcomes in these schools. Indeed, while research points to positive outcomes, there is little depth to the studies. Whilst the individuals teaching the curriculum are growing their perspectives and understanding of the bilingual programmes, they are ideally placed to provide a rich knowledge base surrounding the nuances of the children in this type of education. Moreover, considering the distinctive framework of

bilingual education in England, where English retains its dominant position as the prevailing first language, this study aims to examine these perspectives, acknowledging the outcomes and objectives of these schools as well as the individuals who attend them. This differs from bilingual educational practices in other nations where schools offer disparate goals. Indeed Bialystok (2018) [15] maintains educational systems and attitudes to language will vary internationally, thus rendering context crucial for generalisation and clarity in understanding outcomes.

The free school chosen for this study provides a platform for exploring bilingual education in an English-speaking context throughout the entirety of a child's compulsory schooling. The school aims to nurture bilingualism, biliteracy, and biculturalism in its pupils, primarily through a 50:50 immersion programme, organised by time. Children are taught half the week in English and half the week in their allocated streamed language. As the school is a three-form entry, these languages are Spanish, French, and German, with one class dedicated to each streamed language. Opening in 2012, just two years after the free school initiative was developed, the school selected now teaches from reception year through to sixth form, where students follow the international baccalaureate (IB). Due to its location, pupils are typically from higher SES backgrounds with highly educated parents/carers and come from a range of nationalities/linguistic backgrounds. Children do not necessarily study in their heritage language; a range of families attend with some children studying in their second and third languages (English and their streamed language), others may be studying in their heritage language (streamed language plus English), and others may only have English at home. Furthermore, while parents/carers may request which language they would like their child to study in, they may not necessarily be entered into that class according to capacity; others still may choose a different language to one their child is exposed to at home. Linguistic statistics for attendees of the school are broad and varied, highlighting the nature of bilingual education in this setting. As a result, children may or may not have home support in their streamed language, yet the school aims to ensure the same outcomes for all students who attend. As a free school, the school does not need to follow the national curriculum but is still subject to OFSTED inspections, thus, UK statutory requirements are met through the British curriculum, with teaching being augmented using immersion teaching in the streamed language.

By exploring the experiences and perspectives of teachers, this study, therefore, aims to address the existing gap in the literature by providing qualitative insights into the nuanced aspects of bilingual education and the academic and cognitive effects of bilingual education, including the development of biliteracy skills.

This research addresses three overarching questions:

(RQ1) How do primary school teachers at a bilingual free school describe the children who attend?

(RQ2) How do they perceive the influence of bilingual learning on children's development?

(RQ3) Have they noticed any other effects of bilingual education?

In an increasingly globalised world where written and verbal communication in multiple languages is highly valuable, it is crucial for bilingual education institutions to ensure the best possible outcomes for students regardless of their socio-economic status (SES), culture, or additional educational requirements. This study goes beyond academic achievements and also explores the wellbeing and inclusion of students.

## 1.1. Language Acquisition Theories and the Threshold Theory

Development of the 50:50 immersion programmes discussed in the present study stems from hypotheses and theories surrounding language acquisition. One of the most documented and supported is The Thresholds Theory, hypothesised by Cummins (1976) and Skutnabb-Kangas (1977) (in Baker & Wright, 2017 p. 159) [5] demonstrating the relationship between bilingualism and cognition. Thresholds Theory suggests thresholds, or levels, of linguistic competence an individual must achieve to first, avoid negative impacts

and then to benefit cognitively from bilingualism. Recent support for Thresholds Theory comes from Grundy and Timmer (2019) [20], Paap and Greenberg (2013) [21] and Paap et al. (2014) [22]. Paap et al. [22] challenged the belief that bilingualism offers superior executive function (EF) to that of monolinguals, evidencing that EF advantages from bilingualism may be limited to individuals with greater linguistic proficiency in both languages. Prior to this, Paap and Greenberg (2013) [21] suggested that the advantage may be due to executive control tasks, arguing that the Threshold Theory is a more plausible explanation for observed differences in executive processing between bilingual and monolingual individuals. Finally, Grundy and Timmer (2019)'s [20] meta-analysis uncovered greater working memory capacity in highly proficient bilingual individuals than in lower proficiency and monolingual individuals. They attributed this result to the capacity required to manage two competing languages, eventually giving rise to greater working memory capacity [20]. Criticism for the theory, however, has since led to theoretical evolution, Lehtonen et al. (2018)'s [23] meta-analytic review suggested the relationship between bilingualism and EF is more complex and nuanced than simple thresholds. They concluded that bilingualism is associated with some advantages in certain aspects of EF; however, age and bilingual experience have an effect on the already small effect sizes.

### *1.2. Academic Outcomes of Bilingual Education*

Much of the prior literature on the outcomes of bilingual education has focused on quantitative measures of children's academic outcomes, with a considerable proportion of the research being undertaken in the USA and a smaller but also significant proportion in European countries. Variation in attitudes to language and educational goals according to national and local focus renders generalisation of prior research difficult [15]. However, it is nonetheless prudent to examine studies as educational and social outcomes may still be relevant to the outcomes of pupils in English bilingual schools. Garcia-Centena et al. (2020) [24] in a quantitative study, compared records of monolingual and bilingual schools in Madrid, Spain. Aiming to analyse the impact of bilingual education on academic outcomes, they also considered other variables, including SES. Their results confirmed children in bilingual education did not show reduced performance in subjects taught in either language, most academic results showed parity, and indeed, English results showed significantly greater achievements than in monolingual schools. However, it is important to note that their findings cannot be generalised as other areas and institutions lacked sufficient comparative data. Steele et al. (2017) [25] investigated the effects of language immersion programmes on the test scores of students in the USA, uncovering improvements in literacy performance. There was no detriment nor improvement in maths or science results, however, despite prior research asserting beneficial cognitive effects stemming from greater attention control and improved working memory [25]. Their study offered longitudinal, causal results and included native English speakers as well as those with English as an additional language, with both groups showing similar effects. However, participant results stemmed from families who had applied to a language lottery highlighting increased motivation for language immersion. Results may have varied in a study using participants with reduced motivation to learn. Nicolay et al. (2013) [26] researched 106 eight-year-old children enrolled in either monolingual French classes or immersion English/French classes from the age of five. Assessing attentional and executive functions, the study found significantly faster reaction times from the bilingual group in almost all areas. Children were matched for SES; however, the authors note that a further longitudinal study with more proficient L2 speakers after a greater time in the immersion programme may offer deeper insight.

### *1.3. Non-Academic Outcomes of Bilingual Education*

In 2018, Rona and Mclaughlan [27] investigated New Zealand children studying in three language streams; bilingual, mainstream, and immersive Māori. Results showed the importance of building relationships, effective teaching strategies, and assessment ap-

proaches demonstrating a need for a deeper understanding of biliteracy programmes and appropriate bilingual assessment tools. Indeed, Baker and Wright (2017) [5] highlight the societal importance of biliteracy in accessing culture and traditions, education and enjoyment, and as reinforcement of transmitting oral information. In Romania in 2021, Slapec [28], interviewed bilingual schoolteachers aiming to understand how global competency and knowledge were taught and advanced, considering limited resources. Slapec discovered teachers required greater resources, more partnerships, and more enhanced training and guidance on how to approach global issues. While these studies were small scale and with few participants, they nonetheless outline the importance of in-depth bilingual teacher training, and readily available shared resources. Furthermore, Baker and Wright [5] affirm the importance of such resources, noting positive and varied cultural and societal global outlooks, as well as cognitive advantage, are often attained through biliteracy.

### 1.4. Special Educational Needs in Bilingual Education

In the UK and Ireland, bilingual education previously focused on alternate outcomes, primarily maintenance of an endangered language [5]. Interestingly, love of languages is not the only reason parents select bilingual education. In Ireland, Nic Aindriú (2022) [29] explain, motivation for selecting bilingual education for their children with Special Educational Needs (SEN) may stem from perceived inclusivity. Within the UK, Howard et al. (2021) [7] noted Welsh and English teachers' concerns about the suitability of bilingual education for SEN children, who were concerned about negative developmental outcomes; despite unbalanced participant numbers, similar results were noted across other European studies. In a Swedish survey, Goransson et al. (2022) [30] concluded teachers significantly perceived bilingual education to be at odds with inclusive practices. However, researchers such as Beauchamp et al. (2020) [31], Beauchamp and MacLeod (2017) [32], and Genesse and Lindholm-Leary (2020) [33] examined young SEN learners, concluding they had capacity to learn with no detrimental developmental effects. Furthermore, Beauchamp and MacLeod [32] actually asserted detrimental effects on bilingual children with autistic spectrum disorder when raised monolingually.

### 1.5. Bilingual Education in the UK and Research Gaps

Very few studies to date have focused on bilingual education in England. Saville (2017)'s [10] focus centres around planning and development of bilingual free schools as new institutional developments rather than on the outcomes of the children, while Meier (2012) [34] centres on a London collaboration between a French private "Ecole" and a traditional state school. The 2013 merge created a third bilingual "stream" using a 50:50 immersion pedagogy. Following a mixed methods analysis, results showed improved working relationships, increase in pupil attainment, diverse social populations, multicultural acceptance, and improved academic achievement; however, it was also highlighted that homework and parental involvement were key factors in those outcomes.

### 1.6. Summary, Rationale, and Research Questions

In summary, there is indeed a wealth of recent literature championing the benefits of bilingual education. Baker and Wright (2017) [5] are very thorough in supporting bilingualism and bilingual education; theories discussed are upheld by research such as that undertaken by Grundy and Timmer (2019); Paap et al. (2014), and Paap and Greenberg (2013) [20–22]. Cognitive and academic benefits such as those seen by Garcia-Centena et al. (2020) [24]; Lorenzo et al. (2020) [19], and Gillet et al. (2020) [35] show that those benefits may not be affected by SES and that outcomes improve longitudinally. Finally, the non-academic benefits such as improved relationships [27,34], greater global awareness [28], and inclusive practice benefits for SEN students [31–33] are important socially, culturally, and inclusively. After ten years of bilingual free schools in England, research is still extremely limited. While the academic and cognitive benefits are clear, the social and cultural landscape in England differs from certain regions in Europe and,

indeed, the USA. As such, there is clear scope for understanding the characteristics of those children attending these schools, the impact of this type of education on children raised in England, and other factors experienced by the educators teaching these programmes and an examination of the nuanced factors of those results. This study aims to delve more deeply into teachers' perspectives of the outcomes of pupils in these schools.

## 2. Methodology

### 2.1. Design

This study aims to explore outcomes for children attending bilingual free schools in England through the perspective of educators. Little qualitative research has been compiled on these schools due to their recency and scarcity. A reflexive thematic analysis (RTA) was necessary to gain a more thorough understanding. Braun and Clarke (2022) [36] have refined and demarcated some of the variations on their Thematic Analysis approach, which have emerged over the years, namely, coding reliability, codebook, and RTA [36]. The reflexive approach reflects the researcher's interpretative analysis and indeed emphasises the active role of that researcher [37]. Epistemologically, this research was conducted from a critical realist approach, seeking to understand the subjective reality experienced by those teaching in bilingual education and the social structures that give rise to those phenomena. Ontologically, the data has been analysed from the point of view of a social constructivist paradigm (i.e., in terms of our perspective on reality), recognising the socially constructed nature of knowledge and understanding influenced by social interactions and the social context of teachers in their current roles, past roles and within the broader bilingual school community.

### 2.2. Sample and Procedures

The thematic analysis was conducted at a bilingual free school in an affluent area in the South of England. The school is located in a remote area, which is difficult to reach using public transport but within commuting distance of Oxford, resulting in a high number of Oxford-educated families attending. There are few bilingual schools in England, each unique in their differing pedagogies and outlook. The school in the present study teaches three languages, employing a 50:50 immersion pedagogy divided by time; half the week, teaching the curriculum in streamed language (Spanish, French, or German) and half in English. The school is also the only bilingual free school to continue student's bilingual education through to the secondary level on the same site, where a third or even fourth language can be studied through an immersive 50:50 pedagogy. Attendees of the school tend to come from higher SES status families, and mostly international or with one parent from another country. It is important to recognise that qualitative research should not focus heavily on participant number "*n*" for power of analysis. The Fugard and Potts (2015) [38] model assists in computing optimum sample sizes to best uncover prevalent themes. The present study, therefore, focused on six participants from a variety of backgrounds. Participants had bilingual education teaching experience between 2 and 10 years, ranging in age from 28 to 57, were all female, and both were either bilingual or had conversational skills in two or more languages. They taught a range of year groups from reception to year six, with some having taught in state primary or secondary schools previously, while one had taught in an independent school, one had taught in further education, and most had taught MFL before. Three of the participants taught the Spanish stream and three taught the English side of the French and German streams, providing a balanced overview. See Table 1 for a summary of participant characteristics.

**Table 1.** Participants' biographical information.

| Pseudonym | Age | Year Group Teaching | Teaching Language | Years in Bilingual Education | Years in Traditional Schooling |
|---|---|---|---|---|---|
| Sara | 57 | Y3 | English in French stream | 11 | 7 |
| Natalia | 28 | Reception | English in German stream | <2 | 4 |
| Ingrid | 49 | Y5 | English in French stream | 10 | 8 |
| Margarita | 32 | Y4 | Streamed Spanish | 4 | 2 |
| Collette | 47 | Y2 | Streamed Spanish | 10 | 5 |
| Estelle | 46 | Y1 | Streamed Spanish | 12 | 3 |

An extensive targeted campaign was necessary to reach participants [39]. A two-part sampling strategy was employed as recommended by Drisko (2005) [40], with senior teachers emailing adverts to garner interest alongside a social media campaign. The first participant was purposively sampled, after which snowballing techniques were exercised; this non-probability sampling method aided recruitment by using participants to recruit other participants. After the initial contact, participant information sheets and consent forms were sent, and interview schedules organised. Interviews were conducted primarily using Zoom video conferencing software (Zoom Video Communications, v5.16.2) [41] using closed captions and recordings to aid post-interview transcription. One participant favoured a WhatsApp video call recorded using a second device. Questions were open-ended and follow-up points were included when necessary. Two interviews were interrupted by poor signal, but any potentially corrupted data was repeated to ensure clarity. Interviews lasted between 30 and 70 min, followed by debrief information. All identifying information was removed during transcription.

*2.3. Materials/Measures*

A semi-structured interview was drafted (see Appendix A), allowing for flexibility in the discussion, ensuring that participants from varying backgrounds were given space to fully explore their experiences, and for the researcher to be able to delve more deeply into unexpected responses to elicit more information. Twenty questions were scripted, beginning with seven demographic questions followed by five questions designed to investigate characteristics of children taught, including SES, heritage language, and motivation. The next five questions focused on academic progress, and the final three questions covered other aspects of bilingual education, such as minority groups, casual/playground language, and impact on teachers' language abilities. These questions were developed and inspired by the literature review and the areas on which it was therefore deemed necessary to focus.

*2.4. Data Analysis*

Analysis followed Braun and Clarke's (2006) [42] six-phase process. The first phase, familiarisation with the data, began with transcription using MS Office (version 16.79.1), reading several times and noting down initial thoughts. Phase two saw the generation of initial codes using a three-column table with the transcript central, codes in the right column, and a space on the left for initial theming ideas in phase three. All codes were later transferred to a second document and reordered into groups according to similarity, opposition, and relevance to early theming ideas, and combined across participants for further exploration. Phase four consisted of reviewing the themes and reordering to maintain the integrity of the data without losing depth. The fifth phase of defining and naming themes took much consideration, and finally, the report was produced assisted by highlighted sections of text with compelling evidence and key phrases used to emphasise identified themes.

*2.5. Reflexivity*

It is important to recognise any potential bias that may have been introduced into this study stemming from my position as a parent to children in bilingual education and

from the point of view of a bilingual school governor. Indeed, this may have introduced potential favourable bias into the analysis due to my opinion on bilingual education and the experiences I have had as a parent. This was minimised by maintaining a critically reflexive position, and regularly revisiting the data to ensure the viewpoints examined were clearly defined by participants rather than distorted by my experience and beliefs. Reflexivity was maintained throughout, reflecting on best practice and the clear and implied messages being interpreted by the participants. This also impacted the interview process, and it should be noted that a concerted effort was made to aim for a neutral yet warm, welcoming approach. Indeed, the researcher's positioning in the analytic process fits into the range of epistemological and ontological paradigms within which RTA sits. Furthermore, all but one of the teachers interviewed were bilingual. The only monolingual teacher had some level of ability in several other languages, as well as a deep interest in language learning. Thus, all participants believe strongly in bilingual education and the outcomes for their students.

## 3. Results

This study set out to investigate the impact of bilingual education in England from the perspectives of the educators. Four overarching themes were developed from the data. While most participants agreed across themes, some individual differences arose stemming from their personal backgrounds and experiences. The themes generated were: (1) Academic and Socio-Cultural Effects of Bilingual Education, (2) Privilege in Bilingual Education, (3) Bilingual Education Takes Time, and (4) Special Educational Needs. These will be discussed in turn.

### 3.1. Academic and Socio-Cultural Effects of Bilingual Education

This theme reveals teachers' perceptions of both the academic benefits and the socio-cultural benefits of bilingual education in their experience, particularly in comparison to MFL learning. The benefits of bilingual education do not end with linguistic ability but also, in biliteracy, the ability to read and write proficiently in more than one language.

Teachers noted benefits in language speaking, cognition, and literacy when learning in bilingual education compared to MFL learning, as well as academic and socio-cultural advantages. Indeed, Dosi et al. (2016) [43] highlight that educational settings may play a role in aiding biliteracy and bilingualism, resulting in improved cognitive and linguistic ability. These findings have been further examined in other studies (Cobo-Lewis et al., 2002; Oller & Eilers, 2002, in Dosi et al., 2016) [44]. Furthermore, Tsimpli (2017) [45] maintains that biliteracy and bilingual education positively impact cognitive ability and literacy in additional languages.

Interestingly, teachers placed greater emphasis on the culturally beneficial aspects of bilingual education than on the academic outcomes.

Most of the teachers were experienced in MFL, and they clearly stated the differences between the methods of language teaching and explained why they felt bilingual education was more beneficial. Learning a language through immersion was seen as a much more effective way of learning a language:

Natalia: "*I think, it's just almost the kindest way to learn a language, and also the most erm, the one that can be most adapted*".

Natalia had undergone bilingual education and was multilingual and multiliterate by the time she left school; however, she struggled with Arabic at university due to the teaching methodology which was closer to MFL instruction. Later, in teaching both MFL and the 50:50 immersion programme, Natalia has a deeper understanding of the experience of both learning and teaching using these methods. Her perspective was backed up by her colleagues, who agreed that in terms of bilingual and biliteracy outcomes, the bilingual pedagogy is superior to MFL teaching. Indeed, MFL was described as "boring" and "lacking substance". Participants agreed that the bilingual programme works, and that children are typically linguistically advanced and biliterate by KS2.

Academically, bilingual education is seen as advantageous. Indeed, one participant, Ingrid, the school's maths coordinator, confirmed the noticeable beneficial impacts of bilingual education on maths, spelling, and grammar using her time spent teaching in mainstream education as a comparative marker. Participants also agreed that there were positive neurological benefits and cognitive benefits, improved focus, and concentration, as well as advances in core subjects. It is important to note that, due to the qualitative nature of the present study, alongside the known privileged positions of many of the students at this school, this position may introduce some level of bias. Acknowledging the qualitative nature of this study, therefore, participant's observations and comparisons reflect prior research demonstrating cognitive benefits of biliteracy such as metalinguistic awareness (Bialystock 1991; Sulzby & Teale, 1991, in Dosi et al (2016) [43], cognitive flexibility (Miyake et al., 2000 in Bialystok, 2009) [16], and problem-solving skills [44]. Indeed, Tsimpli [44] highlights that it is these higher literacy skills and inference transfer between languages that suggest literacy skills can also transfer between languages.

Great emphasis was also placed on the socio-cultural benefits of bilingual education. One participant explained that children maintain pride in their heritage language and enjoy speaking it with their parents and community networks:

Ingrid: *"You do hear the European languages being spoken as you walk around the site, . . . that helps children, who, perhaps, if they were erm from a French family and were in an English school, the children start to say, 'Oh, I don't want to speak French with you, Mum. . .That doesn't happen, you don't get people embarrassed of their, their heritage, or their culture, or their particularly the language'"*.

Ingrid's understanding of language learning in a bilingual school highlights the differences she and others experienced from traditional state schools where EAL children may develop a reduced desire to speak their heritage language. In bilingual education, children take pride in languages. Ingrid is the only teacher interviewed who is not fully bilingual. While she has a conversational level understanding of other languages and ten years of bilingual education experience, that experience is framed within this one school. Ingrid has eight years of mainstream education experience, which she refers to as a comparison to outcomes. She has two children attending the school, thereby also offering insight as a monolingual parent/carer introducing bilingualism to her children. Other teachers confirmed the excitement children feel when sharing cultural information with their peers and compared that to less positive experiences in traditional state schools. The impacts of this pride in cultural differences are also felt by the families; indeed, participants spoke of emotional moments with family members living in other countries when communication was facilitated as a result of this learning. This highlights one of the key factors raised, which was the school's outlook on cultural diversity and inclusivity that it not only promotes but also attracts.

Ingrid: *"The thing that really i-is fostered in the school is this. . .the citizenship of Europe, and the world. . .because we have such a multicultural multi-faith demographic, then we pick up on special, um, days and special celebrations that the children are taking part in"*.

The multicultural perspective of bilingual education ensures that international knowledge is strong, not only through the multicultural background of the students but also through its teachings. One participant explained the one assessment area in the Early Years Foundation Stages (EYFS), called people and communities, is the only area that every child reaches "expected" in. The global perspective of the children in this school has far-reaching implications for them in terms of social and cultural acceptance. Indeed, Ingrid confirms the socio-cultural benefits reach children from all backgrounds.

*"I was looking at our data about six months ago. . .our minority groups, if-if together would not be minorities. So we've got a few children and a few families of so many, variety of nations and cultures and faiths that actually, together, we all make up [this school]"*.

Later, she continues:

*"Sharing language by immersion teaching means you're sharing culture. . .what it means to have erm, Korean and Moroccan parents. . .everybody has a place to talk about their heritage and culture and that breaks down barriers".*

Three of the teachers interviewed agreed they gained deep personal satisfaction and enjoyment from teaching in their heritage language. Estelle, Collette, and Margarita all teach the Spanish stream. Estelle has taught bilingual education for 12 years and taught MFL in a traditional school for three years. Whilst she came to teaching later in life, she had volunteering experience in language instruction when living in Spain with Muslim non-Spanish speaking communities. Her experience in Spain and in bilingual education in England highlights her understanding that bilingual education broadens cultural interaction and deepens understanding. Collette confirmed that teaching in Spanish while living in the UK helped her maintain her levels of Spanish. Collette speaks three languages, and before her ten years in bilingual education, she spent five years teaching MFL in a secondary school. Finally, Margarita had two years of experience teaching MFL in an independent school before working in bilingual education; before that, she was a sports coach. For Margarita, her time teaching bilingually has made her feel more at ease due to the wider cultural acceptance. In previous roles, Margarita notes there was some discomfort at the narrower cultural social circles, which tended to result in barriers to communication both with other staff members and with students. She confirms:

*"The environment [at this school] is multicultural, so you, um, feel included. . .you shouldn't worry about your accent. . .children make an effort to understand you because we have the patience to understand them as well so the barrier is completely broken".*

Every participant expressed high levels of pride from working at this school; each of them enjoyed the multicultural atmosphere and being able to teach children in two languages. Conversely, however, Natalia noted deeper gratification from helping children from more deprived backgrounds in the maintained state school she also works in. While motivation is lower and children perhaps more resistant, she felt she had more to offer those children. This feeling was reflected across participants; however, all participants reported a strong feeling of community spirit, which they believed aided motivation and, as a result, learning amongst pupils in their bilingual school.

These academic and socio-cultural benefits address RQ2.

*3.2. Privilege in Bilingual Education*

This theme explains the nature of privilege in this bilingual school. Participants clearly discussed that most of the children in attendance come from higher SES backgrounds and that parents are strongly involved in their children's education. The theme of privilege addresses RQ1.

Natalia: *"I think we only have one free school meal student (in my class), which is, it still shocks me to the core".*

It was agreed by all participants that the average socio-economic status of families attending the school was middle to upper-middle class. This was understandably an uncomfortable concept for some participants, as highlighted in the excerpt above, however, participants also explained how that privilege made a positive impact on learning. The situation above was not outstanding; all participants claimed fewer than 10% of the students they were aware of received FSM. Many theorised that the school is an alternative to independent education for some parents. It was also noted that many of the parents were highly educated themselves and held a strong belief in the education system. This is a school chosen for its outcomes and unique pedagogy. All participants agreed that parents are overwhelmingly encouraging of their children's education and provide support at home. This support is vital for the programme to work, and the school requires a close and streamlined partnership between all stakeholders. Additionally, and perhaps due to parental support and involvement, children at this school are highly motivated. It was also noted that those most engaged parents and carers were more likely to guide children with homework. There was some discrepancy on the concept of homework. Streamed language

teachers firmly agreed that homework does make a difference, those teaching the English half of the week were less reassured. When asked whether homework made a difference, Ingrid responded:

"*(deep breath) Erm, Yes? (tentative tone) but, possibly not as much as you'd expect...So, there's—there's definitely, because there's more support at home, maybe, or there's support between the parents...No, I don't see massive differences*".

Participants did agree, however, that reading at home does make a difference. It was not entirely clear why streamed language teachers saw homework as more essential than those teaching the English side; however, there was an indication that those who completed vocabulary homework would be more likely to make greater gains academically. These gains, therefore, may only be noted in language acquisition; hence, they are only noted by those teaching streamed language. This may also be in part due to the nature of the programme, which is discussed further in the next theme. It was also noted that those unable to help linguistically tended to be in a suitable financial position, which could provide access to private tutors. Furthermore, most participants claimed children from a lower SES home had less parental input, potentially attributed to their lower educational level. Despite this, all participants agreed that additional support was offered to more vulnerable children or those from lower socio-economic classes in the school to work towards educational equity within the school.

Minorities at this bilingual school at least, are class-based, rather than ethnicity-, race-, or culture-based. One participant compared her teaching time at the bilingual school to the time she spent in a traditional state school in a deprived area. She explained the difference in outlook towards languages between the attendees of the two schools.

Natalia: "*Languages can almost be seen as a hobby...and I think that's where the kind of class divide...comes in. It's just, it's like instruments, it's like music, you know? It's beautiful skill, it's people's entire lives, but I don't play an instrument, and I feel nothing is lacking from my life, you know? It's only when you experience it that you think, 'Oh, my goodness! How can people live without playing the violin?'...But it is completely understandable to me. That, that's a thing, a—and I don't know how it could be fixed or remedied, and it, that's really hard to deal with*".

Her final comment indicates a resignation, stemming from frustration in her inability to have an effect; this participant confirmed her preference for teaching bilingual education. However, her conflict arose from the feeling that she had had a greater impact on the children in the traditional state school. She perceives an inequality between classes and demonstrates a deep understanding of why language is not relevant to some.

*3.3. Bilingual Education Takes Time*

This theme discusses the concept of bilingual education as a "buy-in". All participants agreed that to learn bilingually will show positive results but that those results take time. This theme partly answers RQ3.

Natalia: "*Our key stage one. SATs are lower than average, and then our key stage 2 SATs are above average, and there's a real sense of, they need time, we're not giving targets because you will want the vast majority of them to do well and bilingual learners will take longer, and that is fine, and we don't pressure children*".

Natalia explained that in bilingual education, children need more time to reach expected academic standards; this is reflected in the standard assessment tests (SATs) taken in year two (for KS1) and year six (for KS2), which measure a school's academic attainment in maths, reading, grammar, punctuation, and spelling. Interestingly, Natalia also noted that while the school's SAT scores at KS1 were below the national average, they were still above the average of the traditional state school where she also teaches. This state school is in a deprived area, perhaps highlighting further the impact of privilege.

Participants regularly talked about a "buy-in"; this is a free school, but parents are buying into a concept, and strong communication is vital for shared understanding. Participants confirmed if parents understand that bilingualism and biliteracy take time, and that a busy curriculum leaves little time to spare, support is maintained at home, resulting

in better outcomes. This pedagogy requires a week's worth of curriculum taught in half a week to allow the second half of the week to be taught in L2. It was agreed that this results in a tightly packed schedule with no time to cover subjects in more depth. While it can be argued that the repetition of the curriculum in L2 can help consolidate the children's learning in some subjects, the range of teaching methods and autonomy for teachers in the curriculum can vary for other subjects. Ingrid confirms:

"*For example, they do the Tudors in year 5 in English but they do the Middle Ages in French or German and the Spanish do a different period of time so that it fits with something to do with Spain. . .to-to make it culturally relevant*".

Indeed, if children fall behind, it may be difficult for them to catch up due to the pace of the programme, thus, good communication between the school and parents to highlight the time taken is essential. Sara, who is bilingual and speaks two other languages well, has two children who attended the school before it became a bilingual free-school. At that time, the children were taught an hour in their streamed language (not one of the languages Sara's family used at home) a day, and Sara confirmed the difference in how children develop their bilingualism and biliteracy through a 50:50 immersion programme over an hour a day:

"*I can really see a difference in the way we prepare children, when they go into secondary their [stream] language is very strong*".

Sara taught MFL at tertiary and university level for seven years before training as a primary school teacher. She has taught in bilingual education for eleven years.

The development of bilingualism and biliteracy is a gradual process that can take many years; indeed, Gillet et al. (2020) [35] note the advantages in terms of cognitive flexibility and working memory tasks in many cases may take up to six years to emerge, whilst Jared et al. (2011) [45] examined biliteracy predictors over their four-year study confirming the longitudinal aspect of developing biliteracy.

*3.4. Special Education Needs*

The theme of special educational needs (SEN) is important because there were no interview questions based on SEN, yet it was a topic that was raised by all participants. This theme discusses participants' experience of SEN children in bilingual education, equity, and their perception of SEN children's experiences and answers RQ3.

Participants all agreed that children in the school are predominantly motivated, however, where motivation and participation is reduced, it was understood that the most common underlying factor was SEN. It was confirmed that the school caters to high numbers of SEN children, and there was consensus on equity in bilingual education; however, two participants clarified that in certain cases, the children's mental health was more important. At the stage where mental health becomes negatively impacted, parents should consider education with more mainstream pedagogies. Natalia explained that feelings across the school do not always reflect the views of the participants in this study:

"*It isn't necessarily a model that—that works a hundred percent for everyone. . .this school should be right for any child and there's teachers. . .who are very much like, 'No, this isn't for them'. . .and I think that comes from in, France and Spain and Germany, erm there's more special schools for—for needs that aren't extreme, you know there's special schools for dyslexia students, there special schools for ADHD. There's, there's so much provision that I think they-they never necessarily had to deal with it before, and so for them they struggle. . .and actually, for a lot of SEN students the language bit is the only bit they're good at. . .They're good at speaking it, good at understanding it, okay, they can't get it written down necessarily, but don't take away the thing that skill that. . .they enjoy*".

There was a strong communal belief in an inclusive bilingual environment with adapted resources for all children to assist those who require those strategies to function academically, additionally resulting in equity. The above excerpt explains some teachers' insistence that SEN children should not access bilingual education. Interestingly, while she vehemently disagreed, she understood that those teachers' perspectives do not appear

to stem from prejudice but rather from their limited exposure and experience in working alongside children with additional needs. Sara agreed that in some circumstances, SEN may result in difficulties for some children, and participants noted that occasionally if a child's mental health is becoming affected by their struggle with bilingual learning, difficult discussions may require a focus on moving from a bilingual to a monolingual setting.

With regard to the residual skepticism surrounding bilingual education, research illustrates that while bilingual education's cognitive benefits are supported by neuroscientific evidence indicating changes in the brain's executive control networks, this does not consistently manifest as improved behaviour [46]. This divergence underscores the complex interplay between teaching practices and learner outcomes in bilingual contexts. Furthermore, it points to the necessity of considering teacher education and involvement in fostering a nuanced understanding of bilingualism beyond binary classifications toward a continuum of language experiences. This perspective could enrich our approach to educational research and practice.

## 4. Discussion

The themes above highlighted the topics teachers felt were most important in their day-to-day teaching. The first theme of academic and socio-cultural benefits addresses RQ2: the influence of bilingual learning on children's development. Fundamentally it was agreed by five of the six participants who had all taught MFL that bilingual education was superior for language acquisition, supporting Baker and Wright's (2017) [5] classification of weak and strong forms of bilingual education. Baker and Wright state that approaches such as mainstream MFL teaching (a weak form of bilingual education) provide limited enrichment and limited bilingualism, while 50:50 immersion (a strong form of bilingual education) typically results in pluralism and enrichment with biliteracy and bilingualism as outcomes. Furthermore, advances in cognition, focus, and longer-term enhanced results in core subjects were supported as outcomes by participants, upholding hypotheses made by Gillet et al. (2020) [35], which highlighted the attentional and EF benefits of children educated bilingually as well as noting the longer time frame taken to achieve biliteracy. In terms of socio-cultural benefits, this study clearly presents that exposure to a multicultural cohort enhanced by bilingual learning can help shape a child's sense of belonging to a wider multicultural society [12].

The second theme of privilege is important for the future of bilingual education in England and addresses RQ1: explaining how teachers perceive the students within bilingual education. Meier (2012) [34] reported a post-collaborative change, following the merging of two schools, resulting in a bilingual stream, as attracting a higher SES demographic, leading to improved outcomes. This highlights a potentially elitist view of bilingual education whilst noting the potential for aspirant parents to go to the school for social capital [47]. Indeed, the school in the present study may attract higher SES pupils for this reason. The difficult-to-reach location of the school is understood to be a factor in attracting individuals who strongly believe in this specific educational model. Additionally, parents are more able and willing to be involved in their children's education. Some participants shared the view that less well-educated parents do not support their children, and neither do those who have a higher education. Tan et al. (2020) [48] concluded that parental involvement does indeed have a strong impact on student outcomes, compounded by the SES of the parent. The current analysis supported the findings of Tan and colleagues, and participants all noted that parental aspirations had a clear impact on pupil motivation, which was high; however, all participants agreed that their role as educators ensured less supported children were given additional support in school. The relevance of homework led to disagreement among participants. This discrepancy in outlook may stem from the cultural backgrounds and experiences of the streamed language teachers, all of whom are Spanish. Jerrim et al. (2019) [49] note that while teachers in Spain commonly strongly support the importance of homework, no perceived benefits resulted from their longitudinal study. It was clarified several times that those from lower SES backgrounds, while few, were

extremely well supported and cared for and that efforts were underway to improve their outcomes. Nonetheless, as highlighted by Goris et al. (2019) [50] further research should examine the causal effects of SES on bilingual education.

The third theme, Bilingual Education takes time, describes RQ3 and other effects of bilingual education. When assessing a student's attainment, teachers are expected to mark children as working towards, at, or beyond (sometimes called working at greater depth) expected standards. These scores may be based on statutory SATs or in-school assessment methods. Children learning bilingually are almost universally accepted as requiring more time to excel [35] (Thomas & Collier, 1995, 1997, 2002a, 2002b in Baker & Wright, 2017 p. 270) [5], a factor that was understood and explained by all participants. The pedagogy employed in the present study leaves scant time for children who may require extra processing time or who have longer-term absences. Cognitively, bilingual learning is a process, as Baker and Wright (2017 p. 160) [5] note thresholds theory regards bilingualism as requiring levels on which each language builds to achieve or maintain competence in each additional language. One must have a certain level of age-appropriate competence in each language before cognitively progressing to the next level. Educationally this means that plateaus may be encountered while children process each language before progressing as reflected in SATs results; however, as stated previously, by the end of primary, children are regularly achieving above the national average. However, in the present study, it is important to note that SES may have an impact on academic outcomes. While it is clear bilingual education takes time, it was also heavily stated that good communication and guiding parental expectations were vital. Several participants referred to parents "buying-in" to the programme and it was explicitly stated that for it to work, parental involvement and understanding are crucial.

SEN was one of the factors that arose consistently and which all participants felt strongly about despite not being asked explicitly about it. This theme also then relates to the RQ3 regarding other effects of bilingual education. Recent research [30] concluded a considerable number of teachers in Europe are committed to preserving segregation related to SEN in education. However, literature consistently documents that children with SEN are linguistically capable of keeping up with their peers [31]. Furthermore, for those SEN children coming from bilingual homes, attempts to raise them monolingually could be detrimental to the child and their family [30]). Indeed, Baker and Wright [5], (p.347 and p.355) confirm that not only does bilingualism not cause learning difficulties as was once thought, but also that children with additional needs can and will achieve in two languages as far as their individual abilities will allow. Furthermore, benefits will be greater in bilingual special education than in monolingual special education [5] p. 348). Most participants agreed that those with SEN will and do struggle with some aspects of the learning; nonetheless, it also does not discount the fact that SEN children are still able to make achievements bilingually. Nic Aindriu (2022) [29] highlighted an inclusive learning environment and access to a supportive SEN department that motivated parents to elect bilingual education alongside personal factors such as a love of languages and societal benefits. Participants confirmed that the school environment supports a strong SEN department, which could contribute to the high number of SEN pupils in this bilingual learning environment. In the present study, it was confirmed that this SEN cohort was well supported, and an inclusive environment was worked towards. Indeed, teachers only address the potential for alternative schooling if mental health issues arise from the stress of working in two languages.

### 4.1. Limitations and Future Directions

Due to the unique nature of the school in the present study, much of the findings cannot be generalised across other bilingual free schools in England; additionally, the first of these schools in England are currently only ten years old, so longitudinal assessments should examine longer-term outcomes of those who have undertaken bilingual education in their primary stages. The scarcity of these schools in England also hindered participant

recruitment initially, which may lead to potential bias in future studies if recruitment is not carefully controlled. With 23.8% of pupils across the UK eligible for FSM [8] and so few attending this school, there is a need for further research into schools with greater numbers of pupils from more deprived backgrounds with a focus on determining causal factors and outcomes relating to SES. The difficulty in finding participants resulted in a limitation consisting of a less than favourable sample size; furthermore, another limitation was the method of data collection, which again, due to the scarcity of participants available, resulted in purposive and snowball sampling methods, which has the potential to introduce bias into the study. Since this study was actioned, there has been the development of a Bilingual Education Alliance Network (*Home | beaschools*. (n.d.). Beaschools. https://www.beaschools.com/) accessed 8 February 2024 between bilingual free-schools in England [51]. This network can offer access to Head Teachers and Governors at England's free schools, which has the potential to provide support for further research. Furthermore, the RIPL white paper in primary language [52] highlights a strategic role for further research in this field. Finally, we acknowledge here that despite effortful "reflexivity" practices, the credibility of the findings might have been strengthened further using "response validation" or "data triangulation" techniques.

*4.2. Practical Implications*

This study highlights the need for fostering partnerships between bilingual schools, families, and the wider community to enhance support for students and their development. Policymakers may also find these themes useful in developing or revising policies that promote equitable access to bilingual education. Finally, with more research, the ongoing success of bilingual programmes may be a starting point for change in the structure of MFL teaching and linguistic outcomes across the country; by reintroducing languages as a compulsory GCSE subject, more emphasis may then be placed on the method of linguistic delivery, using native speakers and immersive methods, and promoted at an earlier key stage, to achieve preferable outcomes. Policymakers may look to extend the free school scheme to introduce further Bilingual school programmes in England or perhaps extend the current setting to include a bilingual stream. Indeed, the *Research into Primary Languages* (RIPL) white paper [52] notes the limited inspection of primary languages, urging Ofsted and the DfE to incentivise schools to add breadth and depth of learning to their primary languages programmes. Policy implications may include increased funding to upskill language teachers and primary trainees, provision to include language learning pedagogy for trainee teachers, curriculum planning to increase language exposure and appropriate teaching methods, transitions between primary and secondary education, ensuring secondary language learning can proceed from improved linguistic and literacy levels and the use of digital technology.

**5. Conclusions**

To conclude, the present study clearly demonstrates teachers in bilingual free-schools perceive beneficial academic outcomes for their students. There are also clear linguistic benefits over MFL in the classroom, even at the preliminary stages of education, and in the longer term, the benefits of biliteracy, which can have cognitive advantages. The socio-cultural advantages have beneficial outcomes not only for the children but also for their families and the wider community, with students' global knowledge and cultural awareness being well-recognised factors. Nonetheless, there is privilege in bilingual education, particularly in the school represented in the present study. Parental support stemming from privilege may also be a contributing factor to the successful outcomes of the children, and the importance of homework remains undecided. Parental involvement, however, is crucial to a successful programme, and a "buy-in" is involved. SEN is highlighted as a common underlying factor for those children who fall behind; however, those children are supported in an inclusive environment, and participants feel strongly that bilingual education is available to everyone. The strong enthusiasm of the participants involved in

this study and their dedication to a bilingual education will undoubtedly enhance positive outcomes for the children they teach. Furthermore, students are presented with an opportunity to take pride in language learning and their abilities to perform bilingually. Contrary to MFL teaching in traditional state schools and integration aims for EAL children into a monolingual and monoliterate community, this type of education clearly demonstrates a goal towards a more global integration while maintaining a pride in heritage, culture, and an outward-looking community. The novel aspect of these understudied schools would benefit from a great deal of future research whilst further research should consider the importance of biliteracy as an important outcome of bilingual education.

**Author Contributions:** Conceptualization, E.C. and R.F.; Methodology, E.C. and A.H.; Formal analysis, E.C.; Writing—original draft, E.C., R.F. and A.H.; Writing—review & editing, E.C., R.F. and A.H.; Project administration, E.C. and A.H. All authors have read and agreed to the published version of the manuscript.

**Funding:** This research received no external funding.

**Institutional Review Board Statement:** The study was approved by the Faculty Research Ethics Committee (REC) and adheres with the British Psychological Society's Code of Ethics and Conduct (project ID: P4812; date of approval 15 November 2022).

**Informed Consent Statement:** Informed consent was obtained from all subjects involved in the study.

**Data Availability Statement:** Due to the nature of this research, participants of this study did not agree for their data to be shared publicly, so supporting data is not available.

**Conflicts of Interest:** The authors declare no conflict of interest.

### Appendix A. Interview Schedule

Thank you for agreeing to take part in this research.

As you will know from the Participant Information Sheet and Consent Form, this research focuses on understanding teacher (and parent) perspectives on bilingual education and the influence of this on children's development in the Primary School education stages.

Within the interview, I will refer to this bilingual school and bilingual learning environments; note that by bilingual learning, I am referring to the pedagogy employed in the school (i.e., 50:50 immersion) as opposed to MFL teaching often used in primary and more often secondary settings in the UK.

This interview conversation will be recorded and transcribed for later analysis. However, the responses/data you provide will remain confidential and kept on a device requiring password protection in line with current GDPR laws. You are also reminded here that you do not need to answer all the questions, and you are free to withdraw from the study at any point during or after the interview process (until two weeks after the interview has taken place).

Do you have any questions before we start?

Are you happy for me to continue?

Over-arching themes for discussion:

RQ1: How do primary school teachers at a bilingual free school describe the children that attend? (Main questions 1–5)

RQ2: How do they describe the influence of bilingual learning on children's development? (Main questions 6–10)

RQ3: Have they noticed any other effects of bilingual education? (Main questions 11–13)

Initial Questions:

i. Confirm gender and age of participant.
ii. Have you taught in a traditional setting before a Bilingual setting, and could you tell me a bit about your experience of teaching in a traditional primary setting? (Possible prompts—how long, where).

iii.     Could you tell me a bit about your experience of teaching at the school you teach in now? (possible prompts—how long have you taught here? where, how do you feel about the language/bilingual programme?).

iv.     Which environment do you prefer to work in?

v.     Can you tell me more about your own language abilities (prompts—are you bilingual/fluent? How many languages do you speak? What is your mother tongue? If you are learning another language, can you tell me more about how long you have been learning and your language level now.

vi.     Could you tell me about the setting for your current role? (Possible prompt—how long have you been in your current role).

vii.     What is the age range for the children you currently teach, and which age ranges have you taught?

Main Questions

1.     Can you tell me about the kind of children you teach in a bilingual school with regards to their SES? (Possible prompts—What differences are there academically? wellbeing? Are they happy? Engaged? Any examples?).

2.     How do you feel parents engage (or not) with their children's education and how is your communication with them?

3.     Do you think the language spoken in the children's home influences how they perform in corresponding classes? (IE: If they are from a Spanish speaking home do they engage more in classes taught in Spanish) and can you expand on this?

4.     Are the children you teach motivated to speak Spanish? Can you tell me how you facilitate that and how you help reluctant children feel more motivated?

5.     Having worked in both a traditional classroom setting and in a bilingual setting, what are your thoughts about the differences between the environments? (Will not apply to everyone).

6.     In terms of the children's performance academically, what aspects do you consider having an impact? (Language learning in? languages comfortable with? SES?).

7.     Do you notice differences between those children that continue their learning at home (reading, maths games, other homework, etc) and those that do not.

8.     Do you feel bilingual learning helps or hinders the children develop in other academic areas? Can you tell me why? Any specific examples? Does the native language of the child make a difference?

9.     What areas of development have you observed being supported by bilingual learning? (Possible prompts—think about academic learning, socio-emotional development, friendships, confidence, focus, behaviour, engagement, etc., and also tell me how you arrived at these conclusions).

10.     Are there any areas of development that you feel are hindered in a bilingual setting, have there been any specific challenges you've found due to being in a bilingual setting?

11.     How would you describe interactions with children in minority groups and how does this compare to traditional settings you may have taught in? Does sharing language break down barriers?

12.     How does working with children in a bilingual setting affect your own language abilities?

13.     Can you tell me more about the language used in the playground or casual situations and how it relates to children from English speaking homes other non-English monolingual homes and mixed backgrounds?

As indicated, responses may be further probed as necessary to maximise response data using phrases such as, "Could you tell me more about that?" "What was your opinion of that?" "How do you feel that impacts X?")

Thank you for your contributions and valuable time.

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
