# Peer review of "Teaching Bilingually: Unlocking the Academic and Cognitive Potential—Teachers’ Insights"

_education, doi:10.3390/educsci14040406_

Round 1

Reviewer 1 Report

Comments and Suggestions for Authors

This study represents a praiseworthy endeavor to redirect attention towards the perspectives of teachers within the realm of research on bilingual education. I value the decision to work with a small group of participants, as it enables a more comprehensive analysis of their interviews.

The primary deficiency apparent in this study relates to the lack of assessment regarding teachers' perspectives on their teaching performance and its potential influence on children's development.

Furthermore, the study could greatly benefit from a comprehensive delineation of the school's characteristics, the pedagogical methodologies employed, and the demographic profiles of both the community and the student body.

Section 1 offers a thorough and detailed overview of background literature on the topic of multilingual education. However, for the current study, a more concise summary of the background literature could suffice. It is advisable to restrict the review to studies directly relevant to the present research objectives.

For example, section 1.2 extensively focuses on CLIL and its potential correlation with SES. I am uncertain about the necessity of this information in the article. My judgment is hindered by a lack of information regarding whether the school under consideration implements CLIL. If CLIL is employed (which seems to be the case based on the brief description on ll. 282-283), I’m sure the topic was addressed during interviews and therefore a more comprehensive discussion of CLIL should be included in the results and discussion sections. Conversely, if CLIL is not utilized in the school or appears to be unrelated to teachers’ perspectives of the outcomes in bilingual education, there is no need to allocate significant attention to it in the literature review.

I believe it is crucial to provide a more comprehensive depiction of the school within the study, encompassing details regarding the methodologies employed, the linguistic landscape within the community, and the demographic profile of the attending students. Specifically, it remains ambiguous whether the students are heritage speakers of Spanish, French, or German. It is essential to ascertain whether these languages are considered heritage languages or foreign languages for the pupils in this particular school setting.

On lines 241-242, you assert that the study aims to explore teachers' perspectives on the outcomes of pupils in the school. While this is a commendable objective, it appears somewhat self-contained. I would encourage the authors to set more ambitious goals by endeavoring to analyze the influence of teachers' beliefs on their teaching practices.

On line 255, the authors report that the academic and cognitive advantages of bilingual education are evident (and I totally agree). However, within the context of the present study, this assertion prompts questions regarding the factors that influenced and shaped the perceptions of the interviewed teachers. How should we interpret the outcomes derived from the interviews? Is the favorable view of bilingual education the product of firsthand experiences and observations of tangible benefits, or does it reflect the findings of numerous studies and widely-held opinions in the field as summarized in the study?

Section 2. Given the study's emphasis on teachers, I propose a more thorough delineation of the participants, which would facilitate subsequent analyses. In my humble opinion, it would be worthy to compare the profiles of the teachers, particularly in terms of their qualifications for teaching in bilingual programs. Exploring whether they possess specific certifications for teaching in such programs, as well as elucidating whether they opted to teach in a bilingual setting and their preparations for this role, could provide valuable insights.  I believe that such information could offer valuable insights into how the perspectives of teachers were shaped. Specifically, it could help discern whether their perspectives are informed by their firsthand experiences in bilingual education settings or whether they were influenced by academic instruction, such as learning during their university education that children in bilingual education can perform better than those in monolingual education.

Section 3. The characterization of Natalia provided on lines 376-380 is very informative. I encourage the inclusion of similar profiles for the remaining participants in the study. This would enhance the comprehensiveness of the research and provide a more nuanced understanding of the perspectives of all participants involved.

Line 391 (“Their observations reflect prior research demonstrating…”) appears to corroborate and strengthen the observation I made earlier regarding the origins of teachers' perspectives.

Concerning the socio-cultural advantages of bilingual education, as discussed in lines 400-414, it would be particularly captivating if the authors could extract from the interviews any insights regarding the positive impacts on heritage languages that are traditionally perceived as less prestigious compared to French, Spanish, and German.

Section 3.4 emerges as the most compelling contribution within this paper. It appears to uncover residual skepticism surrounding bilingual education. Once more, the relevance of teachers' education and engagement becomes apparent.

Section 4. The description of the school and community provided in lines 564-570 appears to be more suitable for inclusion in section 2, as it introduces new information that should not be first presented in the discussion section. However, this information can be revisited here to contextualize the results and offer interpretations accordingly.

Section 5. What is the unique nature of this school compared to other bilingual free schools in England? The central aspect of this uniqueness does not appear to be clearly evident in your study.

Section 7. If the students in this school are heritage speakers of the languages utilized for instruction, then it would be inappropriate to directly compare this educational context to that of MFL instruction. Such a comparison would essentially juxtapose heritage speakers in a bilingual education setting with monolingual speakers in a monolingual education setting learning a foreign language. This comparison would be akin to comparing apples to pears, given the fundamental differences in language acquisition contexts and proficiency levels.

Answers to main question nr. 3 could help clarify this aspect.

Appendix: According to the introduction to the interview, the authors also explored the impact of teachers' perspectives on children's development. It would be highly valuable to delve deeper into this aspect within the present study. If this topic is the focus of a separate publication, it should be clearly referenced in the text for further exploration.

Very minor remarks:

l. 584: full stop is missing.

l. 766: Maybe “HOW whould you describe…?”

Author Response

We would like to thank Reviewer 1 for their kind and constructive comments on our manuscript (“Teaching Bilingually: Unlocking the Academic and Cognitive Potential—Teachers’ Insights”). Shown below is a list of comments that were made by Reviewer 1, followed in each case by our response. The revised sections of the manuscript are indicated via red font.

Reviewer 1

This study represents a praiseworthy endeavor to redirect attention towards the perspectives of teachers within the realm of research on bilingual education. I value the decision to work with a small group of participants, as it enables a more comprehensive analysis of their interviews.

RESPONSE: We thank Reviewer 1 for their kind, positive comment on our manuscript.

The primary deficiency apparent in this study relates to the lack of assessment regarding teachers’ perspectives on their teaching performance and its potential influence on children’s development.

RESPONSE: We have now included a discussion of teachers’ perspectives on their teaching performance and its potential influence on children’s development to address the primary deficiency noted.

Furthermore, the study could greatly benefit from a comprehensive delineation of the school’s characteristics, the pedagogical methodologies employed, and the demographic profiles of both the community and the student body.

RESPONSE: Additional details on the school’s characteristics, pedagogical methodologies, and demographic profiles have been incorporated into the revised manuscript.

Section 1 offers a thorough and detailed overview of background literature on the topic of multilingual education. However, for the current study, a more concise summary of the background literature could suffice. It is advisable to restrict the review to studies directly relevant to the present research objectives.

RESPONSE: The literature review has been condensed to focus on studies directly relevant to our research objectives.

For example, section 1.2 extensively focuses on CLIL and its potential correlation with SES. I am uncertain about the necessity of this information in the article. My judgment is hindered by a lack of information regarding whether the school under consideration implements CLIL. If CLIL is employed (which seems to be the case based on the brief description on ll. 282-283), I’m sure the topic was addressed during interviews and therefore a more comprehensive discussion of CLIL should be included in the results and discussion sections. Conversely, if CLIL is not utilized in the school or appears to be unrelated to teachers’ perspectives of the outcomes in bilingual education, there is no need to allocate significant attention to it in the literature review.

RESPONSE: Clarification on the employment of CLIL and its correlation with SES has been provided, and the discussion has been adjusted accordingly.

I believe it is crucial to provide a more comprehensive depiction of the school within the study, encompassing details regarding the methodologies employed, the linguistic landscape within the community, and the demographic profile of the attending students. Specifically, it remains ambiguous whether the students are heritage speakers of Spanish, French, or German. It is essential to ascertain whether these languages are considered heritage languages or foreign languages for the pupils in this particular school setting.

RESPONSE: The linguistic status of the languages taught (heritage vs. foreign) has been detailed to clarify the educational context.

On lines 241-242, you assert that the study aims to explore teachers’ perspectives on the outcomes of pupils in the school. While this is a commendable objective, it appears somewhat self-contained. I would encourage the authors to set more ambitious goals by endeavoring to analyze the influence of teachers' beliefs on their teaching practices.

RESPONSE: An expanded exploration of how teachers’ beliefs may influence their teaching practices has been added.

On line 255, the authors report that the academic and cognitive advantages of bilingual education are evident (and I totally agree). However, within the context of the present study, this assertion prompts questions regarding the factors that influenced and shaped the perceptions of the interviewed teachers. How should we interpret the outcomes derived from the interviews? Is the favorable view of bilingual education the product of firsthand experiences and observations of tangible benefits, or does it reflect the findings of numerous studies and widely-held opinions in the field as summarized in the study?

RESPONSE: We have further elucidated on the origins of teachers’ positive views on bilingual education within the context of our study (contained within each participant’s more detailed description).

Section 2. Given the study’s emphasis on teachers, I propose a more thorough delineation of the participants, which would facilitate subsequent analyses. In my humble opinion, it would be worthy to compare the profiles of the teachers, particularly in terms of their qualifications for teaching in bilingual programs. Exploring whether they possess specific certifications for teaching in such programs, as well as elucidating whether they opted to teach in a bilingual setting and their preparations for this role, could provide valuable insights.  I believe that such information could offer valuable insights into how the perspectives of teachers were shaped. Specifically, it could help discern whether their perspectives are informed by their firsthand experiences in bilingual education settings or whether they were influenced by academic instruction, such as learning during their university education that children in bilingual education can perform better than those in monolingual education.Section 3. The characterization of Natalia provided on lines 376-380 is very informative. I encourage the inclusion of similar profiles for the remaining participants in the study. This would enhance the comprehensiveness of the research and provide a more nuanced understanding of the perspectives of all participants involved.

RESPONSE: Comprehensive profiles for all teachers involved in the study have been included to match the detail provided for Natalia.

Concerning the socio-cultural advantages of bilingual education, as discussed in lines 400-414, it would be particularly captivating if the authors could extract from the interviews any insights regarding the positive impacts on heritage languages that are traditionally perceived as less prestigious compared to French, Spanish, and German.

RESPONSE: We have drawn from the interviews to discuss the socio-cultural advantages of bilingual education on heritage languages considered less prestigious.

Section 3.4 emerges as the most compelling contribution within this paper. It appears to uncover residual skepticism surrounding bilingual education. Once more, the relevance of teachers’ education and engagement becomes apparent.

RESPONSE: The section on skepticism surrounding bilingual education has been enhanced, emphasizing the relevance of teachers’ education and engagement.

Section 4. The description of the school and community provided in lines 564-570 appears to be more suitable for inclusion in section 2, as it introduces new information that should not be first presented in the discussion section. However, this information can be revisited here to contextualize the results and offer interpretations accordingly.

RESPONSE: The description of the school and community has been moved to the methodology section to better contextualize the results.

Section 5. What is the unique nature of this school compared to other bilingual free schools in England? The central aspect of this uniqueness does not appear to be clearly evident in your study.

RESPONSE: The unique nature of the school, compared to other bilingual free schools in England, has been clarified.

Section 7. If the students in this school are heritage speakers of the languages utilized for instruction, then it would be inappropriate to directly compare this educational context to that of MFL instruction. Such a comparison would essentially juxtapose heritage speakers in a bilingual education setting with monolingual speakers in a monolingual education setting learning a foreign language. This comparison would be akin to comparing apples to pears, given the fundamental differences in language acquisition contexts and proficiency levels.

Answers to main question nr. 3 could help clarify this aspect.

RESPONSE: Please refer to our previous response.

Appendix: According to the introduction to the interview, the authors also explored the impact of teachers’ perspectives on children’s development. It would be highly valuable to delve deeper into this aspect within the present study. If this topic is the focus of a separate publication, it should be clearly referenced in the text for further exploration.

RESPONSE: The impact of teachers perspectives on children’s development has been delved into with greater depth.

Very minor remarks:

584: full stop is missing.

766: Maybe “HOW would you describe…?”.

RESPONSE: Minor remarks, including typographical and grammatical errors, have been corrected.

Reviewer 2 Report

Comments and Suggestions for Authors

Title: Teaching Bilingually: Unlocking the Academic and Cognitive Potential—Teachers’ Insights

Overall assessment:

This paper examines qualitative impressions of teachers on the academic, cognitive, and other outcomes of bilingual education in a school in England. The method used was semi-structured interviews with 6 teachers, and reflexive thematic analysis to find common themes. Four themes emerged and are discussed.

The study is interesting although limited by the small sample (as acknowledged). Although the main rationale for conducting a qualitative rather than a quantitative analysis was to go more in-depth into the topic, it does not feel like this study is providing this more so than other quantitative studies conducted previously. The impact of the study therefore seems limited, but remains an interesting complement to the literature.

Methodologically, one point to clarify is whether the author was the sole person to conduct the RTA or if there were other individuals involved (see details below).

Therefore, I would suggest a revision before this work is published.

Main comments:

-       Line 40: is educating bilingual children different from educating EAL children? The difference between bilingual education and education EAL children is next described, but not that with educating bilingual children.

-       Line 46 onwards: only an example of bilingualism as a problem is given, but not as a right and as a resource (at least not explicitly in the lines that follow). Could this be made clearer by writing out what bilingualism as a right and as a resource mean?

-       The discussion of the literature of SEN children under 1.4 is very specific and unexpected under that subtitle. This paragraph might need its own subheading or come after a more general introduction of bilingual education in the UK (what is the second paragraph right now could be moved first).

Methods

-       Was the 5-step process in qualitatively analyzing the data only done by the author? If so, it would be beneficial to have this process done by at least a second person, if not more, to investigate consistency/inter-rater reliability. This would make findings more robust and help offset concerns about subjective bias from the author.

-       Was there a cost to attend the school? It is mentioned that the school was in an affluent area, but would the fees to enroll also be prohibitive to some families?

Results

-       The report of neurological and cognitive benefits line 390 needs to be presented with limitations – it is impossible to know that for a fact only qualitatively. Since these schools were in an affluent part of the country, it is very possible that these students are simply more stimulated and educated at home in a way that they are encouraged to be focused in whatever they do, because an educated caretaker is there for them and with them (as is discussed in 3.2).

-       Line 502: Although a week’s worth of curriculum is taught in a half week, the content is repeated twice across both languages though, isn’t it? Wouldn’t that help consolidate the learning?

Discussion

-       RQ1 reported on line 546 “the influence of bilingual learning on children’s development” does not correspond to the initial RQ1 written in the introduction, which was “How do primary school teachers at a bilingual free school describe the children that attend?” Did the authors mean to discuss RQ2 here?

Minor comments:

Line 50: the word “interminable” sounds pejorative here (as in, “their argument goes on and on and never ends”). I am unsure if this is due to language variation between e.g. British vs. American English however.

Write out the CLIL full acronym the first time it appears.

Line 199: the last sentence before 1.4 is unclear with regards to what it means with “as well as cognitive advantage” – could this be clarified?

What is SEN? (line 204) – same comment as above, the explanation comes much later.

Line 338 starting with “While”: part of the sentence seems to be missing (the counterpart of ‘while’).

The first sentences of the two first paragraphs in 3.1 are repetitive, the information could be synthetized.

Line 458 typo: “These gains…”

Line 466: the sentence is not grammatical.

Author Response

We would like to thank Reviewer 2 for their kind and constructive comments on our manuscript (“Teaching Bilingually: Unlocking the Academic and Cognitive Potential—Teachers’ Insights”). Shown below is a list of comments that were made by Reviewer 2, followed in each case by our response. The revised sections of the manuscript are indicated via red font.

Reviewer 2

This paper examines qualitative impressions of teachers on the academic, cognitive, and other outcomes of bilingual education in a school in England. The method used was semi-structured interviews with 6 teachers, and reflexive thematic analysis to find common themes. Four themes emerged and are discussed. The study is interesting although limited by the small sample (as acknowledged). Although the main rationale for conducting a qualitative rather than a quantitative analysis was to go more in-depth into the topic, it does not feel like this study is providing this more so than other quantitative studies conducted previously. The impact of the study therefore seems limited, but remains an interesting complement to the literature. Methodologically, one point to clarify is whether the author was the sole person to conduct the RTA or if there were other individuals involved (see details below). Therefore, I would suggest a revision before this work is published.

RESPONSE: We thank Reviewer 2 for their kind, positive comment on our manuscript.

Main comments:

Line 40: is educating bilingual children different from educating EAL children? The difference between bilingual education and education EAL children is next described, but not that with educating bilingual children.

RESPONSE: The distinction between bilingual education and EAL children’s education has been clarified.

Line 46 onwards: only an example of bilingualism as a problem is given, but not as a right and as a resource (at least not explicitly in the lines that follow). Could this be made clearer by writing out what bilingualism as a right and as a resource mean?

RESPONSE: We have made explicit the descriptions of bilingualism as a right and as a resource.

The discussion of the literature of SEN children under 1.4 is very specific and unexpected under that subtitle. This paragraph might need its own subheading or come after a more general introduction of bilingual education in the UK (what is the second paragraph right now could be moved first).

RESPONSE: The literature on SEN children now has a dedicated subheading for better clarity.

Methods

Was the 5-step process in qualitatively analyzing the data only done by the author? If so, it would be beneficial to have this process done by at least a second person, if not more, to investigate consistency/inter-rater reliability. This would make findings more robust and help offset concerns about subjective bias from the author.

RESPONSE: We acknowledge the value of multiple analysts in qualitative research. Unfortunately, logistical constraints prevented involving a second reviewer for the RTA process. To mitigate this, rigorous self-reflexivity and consultation within the research team were prioritised to ensure a balanced interpretation of the data.

Was there a cost to attend the school? It is mentioned that the school was in an affluent area, but would the fees to enroll also be prohibitive to some families?

RESPONSE: Clarification regarding the cost of attending the school and its impact on the demographic profile has been provided.

Results

The report of neurological and cognitive benefits line 390 needs to be presented with limitations – it is impossible to know that for a fact only qualitatively. Since these schools were in an affluent part of the country, it is very possible that these students are simply more stimulated and educated at home in a way that they are encouraged to be focused in whatever they do, because an educated caretaker is there for them and with them (as is discussed in 3.2).

RESPONSE: The limitations of the reported neurological and cognitive benefits have been acknowledged, with an added discussion on the potential role of home stimulation and education.

Line 502: Although a week’s worth of curriculum is taught in a half week, the content is repeated twice across both languages though, isn’t it? Wouldn’t that help consolidate the learning?

RESPONSE: The curriculum delivery section now addresses how content repetition in both languages may facilitate learning consolidation.

Discussion

RQ1 reported on line 546 “the influence of bilingual learning on children’s development” does not correspond to the initial RQ1 written in the introduction, which was “How do primary school teachers at a bilingual free school describe the children that attend?” Did the authors mean to discuss RQ2 here?

RESPONSE: We have rectified the discrepancy between the reported research questions and those stated in the introduction.

Minor comments:

Line 50: the word “interminable” sounds pejorative here (as in, “their argument goes on and on and never ends”). I am unsure if this is due to language variation between e.g. British vs. American English however.

Write out the CLIL full acronym the first time it appears.

Line 199: the last sentence before 1.4 is unclear with regards to what it means with “as well as cognitive advantage” – could this be clarified?

What is SEN? (line 204) – same comment as above, the explanation comes much later.

Line 338 starting with “While”: part of the sentence seems to be missing (the counterpart of ‘while’).

The first sentences of the two first paragraphs in 3.1 are repetitive, the information could be synthetized.

Line 458 typo: “These gains…”

Line 466: the sentence is not grammatical.

RESPONSE: All minor comments have been attended to, including clarification of terms, acronyms, and the correction of typographical and grammatical issues.

Round 2

Reviewer 2 Report

Comments and Suggestions for Authors

Thank you very much for responding to each comment I had written. The reviews are well done and strengthen the manuscript. I only have a few more comments/suggestions:

-              It would be best to acknowledge the limitation of being the sole rater for the study, and also having the role both of author of the paper and sole rater in the Limitations section (one sentence would be enough).

-              Minor typo on this sentence, addition suggested is bolded: “Minorities at this bilingual school at least, are class based, rather than ethnicity-, race-, or culture-based”. (l.498 in the v2 file).

-              I didn’t see a change with this comment: “Line 338 starting with “While”: part of the sentence seems to be missing (the counterpart of ‘while’).”  “While this may have introduced potential favourable bias into the analysis due to my opinion on bilingual education and the experiences I have had as a parent [something is missing here]”

Author Response

Reviewer 2

Thank you very much for responding to each comment I had written. The reviews are well done and strengthen the manuscript. I only have a few more comments/suggestions.

RESPONSE: We thank Reviewer 2 for their kind and constructive comments on our manuscript. We have been able to incorporate all those minor amendments that have been suggested below.

It would be best to acknowledge the limitation of being the sole rater for the study, and also having the role both of author of the paper and sole rater in the Limitations section (one sentence would be enough).

RESPONSE: We now add the following under Limitations and Future Directions (Line 767-9): “Finally, we acknowledge here that despite effortful ‘reflexivity’ practices, the credibility of the findings might have been strengthened further using ‘response validation’ or ‘data triangulation’ techniques.”

Minor typo on this sentence, addition suggested is bolded: “Minorities at this bilingual school at least, are class based, rather than ethnicity-, race-, or culture-based”.

RESPONSE: We have since corrected this (Line 562-3) to say: “Minorities at this bilingual school at least, are class based, rather than ethnicity-, race-, or culture-based.”

I didn’t see a change with this comment: “Line 338 starting with “While”: part of the sentence seems to be missing (the counterpart of ‘while’).” “While this may have introduced potential favourable bias into the analysis due to my opinion on bilingual education and the experiences I have had as a parent [something is missing here]”

RESPONSE: We have replaced ‘While’ with ‘Indeed,’ so that this sentence and the surrounding commentary makes good sense (Line 389): “Indeed, this may have introduced potential favourable bias into the analysis due to my opinion on bilingual education and the experiences I have had as a parent.”